# Developing a recovery-focused therapy for older people with bipolar disorder: a qualitative focus group study

Elizabeth Tyler  , Fiona Lobban, Rita Long, Steven H Jones

Division of Health Research, Lancaster University, Lancaster, UK

**Correspondence to**
Dr Elizabeth Tyler;
e.tyler@lancaster.ac.uk

## ABSTRACT

**Objectives** As awareness of bipolar disorder (BD) increases and the world experiences a rapid ageing of the population, the number of people living with BD in later life is expected to rise substantially. There is no current evidence base for the effectiveness of psychological interventions for older adults with BD. This focus group study explored a number of topics to inform the development and delivery of a recovery-focused therapy (RfT) for older adults with BD.

**Design** A qualitative focus group study.

**Setting** Three focus groups were conducted at a university in the North West of England.

**Participants** Eight people took part in the focus groups; six older adults with BD, one carer and one friend.

**Results** Participant's responses clustered into six themes: (1) health-related and age-related changes in later life, (2) the experience of BD in later life, (3) managing and coping with BD in later life, (4) recovery in later life, (5) seeking helping in the future and (6) adapting RfT for older people.

**Conclusions** Participants reported a range of health-related and age-related changes and strategies to manage their BD. Participants held mixed views about using the term 'recovery' in later life. Participants were in agreement that certain adaptations were needed for delivering RfT for older adults, based on their experience of living with BD in later life. The data collected as part of the focus groups have led to a number of recommendations for delivering RfT for older adults with BD in a randomised controlled trial (Clinical Trial Registration: ISRCTN13875321).

## BACKGROUND

Approximately 25% of individuals living with bipolar disorder (BD) are of age 60 years or over.[1] As awareness of the condition increases and the world experiences a rapid ageing of the population [2], the number of people living with BD into later life is expected to rise substantially. Research indicates that older adult-specific services are better placed to meet the needs of those with mental health problems in later life, when compared with general adult services.[3] The National Institute for Health and Care Excellence BD guideline[4] suggests that older adults with BD should be offered the same treatments as younger people, however there appear to be

### Strengths and limitations of this study

► To our knowledge this is the first study to involve older adults with bipolar disorder (BD) to shape and develop a psychological intervention specifically for their cohort.
► The study was designed and conducted in consultation with service user representation throughout, enhancing the quality, value and the relevance of the study.
► The clinical recommendations for delivering recovery-focused therapy for older adults have been developed in partnership with individuals with lived experience of BD, carers and healthcare professionals.
► The individuals with lived experience taking part in the groups were not representative of all older people with BD. They were all white British and all had (or had retired from) a professional working background.

unique, clinical characteristics that feature in the older population,[5] which may impact on their needs and response to treatment.[6]

Older people with BD follow a chronic and persistent course,[7] with recurrent mood episodes continuing beyond the age of 70 years.[8] The clinical features include poorer cognitive functioning, even during periods of mood stability,[9] which may impact on functioning, leading to problems with finances, domestic roles, mobility and social and recreational activities.[7] Rise *et al*'s[10] systematic review found that older people with BD are more likely to present with conditions such as diabetes mellitus, cancer, thyroid disorder and hypertension compared with age-matched controls. Older people with BD may be twice as likely to experiences stressful life events compared with healthy controls such as changes in familial structure, retirement, housing and finances,[11] which can act as triggers for mood episodes.[7] Studies have found that older people with BD are more likely to experience depressive episodes[12 13] and there

is some evidence to suggest milder episodes of mania, compared with their younger counterparts[14]

At present, there is no evidence base for the effectiveness of psychological interventions for older adults with BD. The aim of the study was to conduct a number of focus groups with people with lived experience of BD in later life, to adapt a recovery-focused therapy (RfT) intervention developed for younger adults,[15] so it could be offered to people over the age of 60 years. The focus group work sits within a larger study which consists of two phases.[16]

The key aims of the focus groups were to: (1) explore the extent to which the original RfT intervention was acceptable to older adults with BD, (2) identify whether any adaptations would be needed to the existing manual and what support older adults would want from a therapist during therapy and (3) explore the experience of BD in later life, including the relationship with relatives and health professionals and the concept of recovery with BD.

## METHOD
### Design
The *British Medical Journal* and the Medical Research Council recommend the incorporation of qualitative research in the process of complex intervention development.[17] [18] Focus group methodology[19] was chosen because the researcher was interested in understanding the topics from a diverse range of perspectives, moderating the discussion from a peripheral role.[20] The Standards for Reporting Qualitative Research were adhered to and a copy of the checklist can be found in online supplemental file 1.

### Patient and public involvement (PPI)
PPI representatives were involved in the study design process from the outset. The team has a service user advisory panel, led at the time by RL. The panel provided feedback on the original grant application and reviewed study documents, including participant information sheets, consent forms and topic guides. At the end of the focus group, participants were invited to remain in the study and form the service user reference group to contribute towards the design, implementation and dissemination at phase two.

### Sampling and recruitment
Participants were recruited via a confidential database of individuals who have previously consented to being approached about potential involvement in research studies. To be eligible to take part in the study, participants were required to identify themselves as a person

| Table 1 | Focus group topics |
|---|---|
| **Focus group** | **Topics explored** |
| 1 | 1. Overview of proposed therapy<br>2. Living with BD in later life<br>3. Experience of recovery in later life |
| 2 | 1. Adapting RfT for older people<br>2. What support people want from a therapist during therapy<br>3. Relationship with relatives and health professionals |
| 3 | 1. All topics revisited |

BD, bipolar disorder; RfT, recovery-focused therapy.

over the age of 60 years living with BD or be a relative or a friend of an older adult with BD (to offer a diverse range of perspectives), have the capacity to provide informed consent and have sufficient English language skills to read the information sheet and take part in the discussions.

We aimed to recruit approximately 6–12 participants, a combination of males and females, to take part in the focus groups, consistent with focus group methodology.[19 21 22] We intended to recruit at least six people with lived experience of BD in later life and additional carers, relatives or friends to broaden the discussion. All interested individuals were invited to attend each of the three focus groups.

### Topic guide for focus groups
A topic guide was developed with the research team (ET, SHJ, FL and RL) and designed to loosely structure the focus group and lead the discussion. See online supplemental file 2 for focus group topics and content. Different topics were explored in groups 1 and 2 (see table 1) and during group 3 the researcher revisited all six topics to gather more rich and detailed information and ensure that all participants taking part had the opportunity to express their ideas on each of the areas. See table 2 for who attended each group.

### Procedure
The groups were conducted at a university in the North West of England and lasted for approximately 90 min, consistent with focus group methodology.[19] The first focus group was facilitated by two members of the research team (a service user researcher and the lead researcher). The service user researcher was not able attend groups 2 and 3 which were facilitated by the lead researcher alone. All groups were audio recorded and transcribed prior to analysis. All participants provided written informed consent.

| Table 2 | Groups attended | | | | | | | |
|---|---|---|---|---|---|---|---|---|
| **Participant number** | **P001** | **P002** | **P003** | **P004** | **P005** | **P006** | **P007** | **P008** |
| Focus group attended | 1 and 2 | 1–3 | 1 | 1 and 2 | 1 and 3 | 1 and 2 | 2 and 3 | 2 |
| Service user/carer/friend | Friend | Service user | Service user | Service user | Service user | Carer | Service user | Service user |

## ANALYSIS

The focus groups were transcribed and analysed thematically using framework analysis,[23] a popular way to analyse primary qualitative data in the area of healthcare.[24] This allows for both deductive and inductive coding, with concepts or themes identified as coding categories a priori to be combined with other themes that emerge de novo.[24] Five topics formed the original framework: living with BD in later life, experience of recovery in later life, adapting RfT for older people, what support people want from a therapist during therapy and the relationship with relatives and health professionals.

Furber[25] identified five phases of framework analysis: familiarisation, a theoretical framework, indexing, charting and synthesis. The three transcripts were read and reread by the lead researcher (ET) to aid the process of familiarisation, before undertaking the initial coding for transcript 1. Two members of the research team (SHJ and FL) independently read and coded (indexed) transcript 1. A meeting took place within the team (ET, SHJ and FL) to discuss the transcript, why the coded sections had been interpreted as meaningful and to discuss new codes and the development of the theoretical framework. The new framework was then applied to transcripts 2 and 3. Team meetings took place to discuss any further amendments to the framework, based on the emergence of new codes. The final framework was a representative of the entire data set collected from the three focus groups.

### Reflexivity

The researcher was aware that they (and other members of the research team) were approaching the study with a set of preconceived ideas about the recovery-focused approach and the direction of therapy. SHJ developed the original recovery-focused CBT manual for an adult BD population. SHJ and FL were involved in a recently completed trial exploring its feasibility and acceptability.[8] Subsequently, ET, SHJ and FL were involved in a case series looking at the application of this approach for individuals with a more established BD diagnosis. The researcher aimed to facilitate the discussion regarding these topics in an open fashion to ensure that the participants felt able to give open and honest feedback about the approach. Similarly, when analysing and interpreting the results, the role of potential bias was highlighted to ensure that the results were a true reflection of the participant's ideas, rather than the team selecting the responses which aligned with their personal view on the topic.

## RESULTS
### Participants

All participants were invited to attend each focus group; however, this was not achievable due to their various commitments and therefore we were flexible in response to their availability. As shown in table 2, eight participants took part across the three focus groups attending 1–3

**Table 3** Participant characteristic for individuals with lived experience

| Participant characteristic | N (6) |
|---|---|
| Ethnicity | |
| White British | 6 |
| Age (years) | |
| 65–70 | 2 |
| 70–75 | 2 |
| 75–80 | 2 |
| Gender | |
| Male | 3 |
| Female | 3 |
| Length of diagnosis (years) | |
| Less than 10 | 1 |
| 10–15 | 1 |
| 15–20 | 1 |
| 20–30 | 2 |
| 30+ | 1 |
| Current occupation | |
| Retired | 5 |
| Director of a company | 1 |
| Occupation prior to retirement | |
| Director of a company | 1 |
| Company secretary | 1 |
| Teacher | 1 |
| University researcher | 1 |
| Nursing | 1 |
| Past experience of psychological therapy | |
| Yes | 4 |
| No | 2 |
| Type of therapy | |
| Cognitive behaviouyral therapy (CBT) | 3 |
| Mindfulness | 1 |

groups each, six service users with lived experience, one carer and one friend (who also had a diagnosis of BD and identified herself as moving into later life). As shown in table 3, participants with lived experience ages ranged from 67 to 77 years (M=72), with an established diagnosis ranging from 8 to 38 years (M=20).

### Themes

There was a great deal of rich data regarding how participants described perceived changes during their later years. Once the team met to code focus group 1, using a combination of both inductive and deductive analysis, they agreed that the framework should be revised to reflect the data. The original topic 'living with BD in later life' was broadened to encompass three themes: 'health-related and age-related changes in later life', 'experience of BD in later life' and 'managing and coping with BD

in later life', following an inductive approach. Initial topics of interest 'relationship with relatives and health professionals' became subthemes within the new theme 'managing and coping with bipolar in later life'. The original topic 'what support people want from a therapist' became a new theme 'seeking helping in the future'. The original topics of interest 'experience of recovery in later life' and 'adapting RfT for older people' became the final themes, following a deductive approach. Please see table 4 for the preliminary and final themes. Additional example quotes can be found in online supplemental file 3.

### Theme 1: Health-related and age-related changes in later life

In general, the group appeared to have experienced a number of changes in later life. These included physical problems (eg, arthritis, back problems and hearing difficulties), cognitive changes (eg, decline in memory, concentration and increased distractibility) and behavioural changes (eg, not playing sport or reading any longer). There was a sense of frustration and sadness about not being able to engage in activities that they had previously enjoyed. What was interesting in this context was how they made sense of these changes in relation to their BD diagnosis. With regard to their difficulties with memory and concentration, the group had trouble identifying whether this was part of the natural process of ageing or caused by their BD.

> Now how much of this is due to I'm getting older; how much is due to bipolar? P002, FG2

This was particularly confusing as many people noticed differences in their memory depending on what mood state they were experiencing. So generally, when they felt low, they felt their memory difficulties worsened, however when their mood was higher some participants reported their memory problems appeared to dissipate.

> I am…on the ball. I can remember anything…As soon as my mood starts to dip then I start to not. I can't remember what I did the day before. P001, FG2

The group reported changes in their family, work and social structure, meaning they had more time on their hands, which for some led to feelings of loneliness.

> I've lost such a lot of good friends…such a lot gone…I miss some of them around me. P004, FG1

While changes in family, work and social structure are commonly reported by older adults,[20] for this group having a life history of BD appeared to have had an additional impact. The group found having more time to themselves and dwelling on events that had happened in the past (often related to situations that had arisen or been exacerbated by their behaviour while in a mood episode), led to feelings of guilt and shame. It was interesting that these stories were relayed with regrets that still felt quite raw, with people

**Table 4** Themes

| Preliminary theme | Final theme | Subtheme |
|---|---|---|
| Living with BD in later life | Health-related and age-related changes in later life | Cognitive symptoms |
| | | Behavioural changes |
| | | Physical symptoms |
| | | Social network changes |
| | | Loneliness and isolation |
| Living with BD in later life | Experience of BD in later life | Change in psychological symptoms |
| | | Identity |
| | | Stigma |
| | | Guilt and shame |
| Living with BD in later life | Managing and coping with BD in later life | Psychological help |
| | | Self-management |
| | | Medication |
| | | Distraction |
| Relationship with relatives and health professionals | | Importance of a meaningful activity |
| | | Help from health professionals |
| | | Help from relative |
| Experience of recovery in later life | Experience of recovery in later life | Person's concept of recovery |
| | | Messages from health professionals about recovery |
| | | Management strategies to achieve recovery |
| | | Gaining a meaningful activity to aid recovery |
| | | The importance of hope |
| Support from the therapist | Seeking help in the future | Building relationship with therapist |
| | | Therapist transparency |
| | | Important goals for therapy |
| | | Building on strengths and resilience |
| | | Harder to seek help |
| | | Time running out to change |
| Adapting RfT for older people | Adapting RfT for older people | Session length |
| | | Memory and learning techniques |
| | | Enhancing the therapeutic experience |
| | | Using clear and simple language |
| | | Making study materials interesting and accessible |
| | | Booster sessions to optimise outcomes |

BD, bipolar disorder; RfT, recovery-focused therapy.

wishing they had behaved differently and treated family members with more respect.

> Coming to terms with events that have happened and you can't understand why that happened and why

you did that….and with all the embarrassment to cope with. P001, FG1

For other individuals in the group, not having to worry about the pressures of working life was seen in a more positive light with regard to the effect on their mood.

I don't have the stress of working life… and probably the stress makes it worse. P007, FG3

### Theme 2: Experience of BD in later life

All participants in the group reported a lifelong history of mood instability. In general, they felt their experience of both mania and depression had changed over the years, although patterns of change varied. What was common to all was a need to make sense of why this change was happening. Some people felt that they had fewer episodes now they were older, whereas others felt they had more but they were shorter and milder. In general, the group felt that they experienced more lows than high mood currently and some individuals felt their depressions had worsened over time compared with when they were younger. With regard to their depression, one participant stated:

I have a knock down every day, two hours. Not heavy but I just close down. P002, FG1

In relation to mania, the majority of the group felt that their episodes were not as intense as they had been previously. The carer in the group described how their relative's mania was less physical and more cognitive than previously:

It wasn't physical…It was more lots of disturbed thoughts. P006, FG1

The impact of living a life with a persistent mental health problem was evident within the group. Over the years they felt they had experienced stigma from society and a change in their social networks.

And people think because you're mentally ill, in inverted commas, that you need to be treated differently. P005, FG3

Some members of the group felt they were now struggling with their sense of identity which had impacted on their confidence and self-esteem.

I think it loses itself because you can't do the things you did… who am I now? What have I got to contribute? P002, FG3

This struggle appeared to be linked to the combined consequences of the ageing process and the repeated episodes over the lifespan where the regrets still felt salient.

I have to say my confidence is zilch now because of the events. P002, FG3

However, not all members of the group were experiencing these problems: one individual described himself as having a very 'solid identity…with that workload' (P007, FG3), which he attributed to the fact that he had a voluntary job. Another female felt she had re-established her identity by going out and mixing with people and building up a community for herself:

I'm a member of various organisations… I've got an activity every single day and I'm mixing with people and this helps enormously. (P005, FG3)

Here, the length of time since diagnosis does not appear to be a factor (P007 had a fairly recent time since diagnosis of 8 years and P005 received her diagnosis 26 years ago). Instead, having a role, a sense of purpose and belonging appears paramount to maintaining and enhancing a person's self-worth and resilience in later life.

### Theme 3: Managing or coping with BD in later life

The group had used a range of strategies over the years to manage their BD, including medication, psychological therapies and self-management strategies. All of the group (apart from the relative) reported that they were taking medication and in general felt it had helped to stabilise their episodes of BD. However, there were concerns about the effects of taking medication on their memory and concentration, which seemed to be intensifying as people were taking the medication over longer and longer periods of time.

Two members of the group had experience of their psychiatrist reducing their medication as they got older. They both felt that they had been overmedicated prior to the reduction.

They saw I was a zombie and I didn't think I was capable of anything. (P005, FG1)

They both felt positive about the care that they had received from their psychiatrist in relation to the medication reduction.

The group acknowledged a change in approach over recent years with regard to psychological therapy and the majority of the group had some experience of psychological therapies (see table 1). A few members wondered about what other older friends with BD might think if offered a psychological approach.

I'm not sure he would find that… a bit weird. A bit like whacky. Because he's nearly sixty, he's never been offered anything…. P001, FG1

Some members of the group felt they had learnt to manage their BD more effectively as they became older. Over the years, they felt they had learnt strategies to self-manage the condition through their own experience of living with BD and reading available information. They felt they were able to stabilise themselves more quickly in response to mood fluctuations, reporting conscious lifestyle changes including looking for triggers.

I read up on it and I learnt all my trigger points; I now identify the illness whereas before I just thought it was the way my life was going. P005, FG1

One participant attributed this change in self-management to her community psychiatric nurse. The enhanced sense of responsibility to manage her own condition appeared a key factor in building the therapeutic relationship. This appeared to be significantly different to the care she had received in the past.

Having a really good CPN who has given me responsibility, and believes in me, has really helped. (P005, FG3)

However, another participants felt that they were still engaged in a struggle with professionals which had persisted over the years. There was a sense of disappointment and frustration regarding the ongoing battle with professionals and an awareness that stigma related to living a life with BD may still be evident within some services.

We're still fighting professionals who don't believe that we are capable of what we truly are. P005, FG1

The group used distraction as a way of coping on a more day-to-day basis, reporting activities such as gardening or voluntary jobs as a way of keeping the mind occupied and not focusing on past events. However, one group member acknowledged how distraction only worked as a strategy in the short term.

If you're doing something that doesn't require a lot of concentration, these things come back to the mind. P002, FG1

There appeared to be a mixture of experiences in relation to the help that individuals had received from relatives and friends to cope with their BD over the years. Maintaining a strong and supportive network of people around the person with BD was highlighted as an important coping factor by a number of group members, particularly as the person reaches his/her later years as roles change (eg, retirement, family structure changes) and loneliness can become apparent.

Participants were able to identify times where they had found input from a relative/ friend helpful:

if I went on a low my partner, my mate, my working partner knew the symptoms and he'd tell me, you know … you've got to be careful… P004, FG1

However, one participant felt that her episodes were triggered by her husband's behaviour and now he was no longer in the home, her mood was stable and she is in a stronger position to cope with her BD episodes.

I no longer have that trigger so I am stable. P005, FG3

## Theme 4: Experience of recovery in later life

The concept of recovery in later life was introduced using the quote from the Scottish Recovery Network (see online supplemental file 1). There appeared to be a division in the group with regard to their concept of 'recovery' in later life and their ability to recover with BD. Some participants questioned if the word recovery felt too final and a word like 'stability' may be more appropriate. These individuals were aligned with the traditional 'clinical' view of recovery and felt that being 'recovered' meant not experiencing any symptoms of BD in the future.

Recovery means you're recovered…. You're cured and bipolar can't be cured… P001, FG1

Recovering with BD felt like an unachievable aim:

Something that I look for beyond the horizon. P002, FG1

In this sense, these participants questioned their ability to 'recover' in later life and to what degree they could move on and change. There appeared to be a link between their perception and messages they had received in the past about recovery.

I've been told that you can never recover from bipolar. P002, FG1

However other members of the group felt they were 'recovered' or were in the process of recovering. One individual described how recovery for the mental health team was the person not going back into hospital, whereas for them it was getting back to work or engaging in a meaningful activity. Therefore, he was able to differentiate between the messages from professionals regarding 'clinical recovery' and their own personal recovery journey.

Participants who were identified with the more personal concept of recovery reported a range of strategies to support their recovery journey; setting a goal and engaging in a meaningful activity felt important, alongside the use of medication and self-management strategies. They acknowledged how important hope was as part of their recovery journey.

You need hope. (P004, FG1)

In general, there was this strong sense of wanting to contribute to society and look for new opportunities following their lifetime of repeated episodes and transition into later life.

It's about setting out the right goals as well…isn't it? (P004, FG1)

Despite having to deal with the combined societal stigma of BD and becoming older, some people could clearly identify benefits from these simultaneous circumstances. There was a sense that they had really 'survived' the struggle with living a life with bipolar and could use this wealth of experience and knowledge to help their younger counterparts.

Our experiences in life have given us knowledge that we can help other people much better…we know what works and what doesn't. (P005, FG1)

Others found it harder to see these benefits and seemed to have their own stigma regarding becoming older and the limitations this puts on opportunities for personal growth and development:

I'm not sure at my age just to what degree I can move on. (P002, FG1)

### Theme 5: Seeking help in the future

Nearly all the members of the group had experience of psychological therapies in the past (see table 1). They were positive about the development of a tailored intervention specifically for their age group and they had some very clear ideas about the sort of relationship they would like with a potential therapist.

They need to have listening skills. They need to be adapting body language and tone of voice and pitch of voice. They need to be emphatic…they need to be aware of the sort of problems we face…be able to get our trust. P005, FG3

They wanted to be treated with dignity and respect, but there was an additional need in this group for there to be a shared understanding of the wealth of experience the older person with BD was bringing to the therapeutic relationship.

They wanted transparency when it came to acknowledging any differences between the therapist and the client with regard to age, gender, ethnicity and the therapist's lack of BD diagnosis. They felt it was very important for the therapist to spend time building a relationship, allowing time at the beginning and the end of the session to really get to know the person and their living history. It was felt more time might be needed for this as their histories were often longer and more complex as a result of their age.

It's about having that personal relationship…It takes two to three sessions to get the confidence. P004, FG2

The group wanted the therapist to work on issues such as assertiveness, confidence building and improving competence, maintaining an encouraging stance throughout. This seemed to be driven by a need for the therapeutic process to help challenge some of the stigmatising views of older adults in our society:

We need loads of encouragement and appreciation that we are valued members of society. P005, FG1

In relation to seeking help, participants reflected on their own experiences and wondered if they would only engage in therapy when they were in an episode, perhaps influenced by negative experiences of care in the past. The carer in the group felt their relative found it harder to seek help now she was older because the repeated episodes over time (and limited life span) had led to a greater sense of hopelessness.

### Theme 6: Adapting RfT for older people

With regard to the specific recovery-focused intervention, the group appeared positive in general about the development of the specific therapy. They felt that the optimum session time was 50 min to an hour. They felt that it was necessary to make some adaptations to the therapy, based on their experiences of living with BD in later life and the changes they had experienced. With regard to memory difficulties, they felt using strategies such as repetition and association would be helpful and writing summaries at the end of each session and revisiting these at the beginning of the next session.

When you get older, you know, with sight problems and hearing problems you just need more resources… things written down… sort of more back up. P008, FG2

They suggested using images, film or audio recordings as a way of enhancing a person's therapeutic experience.

The group emphasised the importance of using clear and simple language and making study materials accessible and visually interesting.

Some large sort of text…if it's complicated, it could be simplified. P006, FG2

The groups were mixed in their opinion of how many sessions would be optimal, some suggesting 6–8 sessions and others thought up to a year was needed. The group suggested that booster sessions would be helpful at the end of therapy to revisit strategies and enhance outcomes.

If you've learnt some practical strategies…especially with CBT…they you've had some time to practice on your own. P008, FG2

### DISCUSSION

The aim of this study was to better understand the experience of living BD, the concept of recovery and explore whether any adaptations were needed to the existing therapy RfT manual. The findings suggest that the groups were experiencing similar health-related and age-related changes compared with other older people living with BD, such as changes in cognition[26] and an increase in physical health conditions.[27 28] The group also appeared to be experiencing the same issues as older people with other mental health issues such as losses, loneliness and isolation.[29] However, the consequence of having more time on their hands in later life (due to a reduction in social networks and loss of roles (eg, work)) presented a specific challenge to this group of people. Time alone offered the opportunity to ruminate on negative events that had happened in the past which appear to be more salient in this group of people due to the nature of BD, raising issues such as guilt and shame. Interestingly, some

participants felt a sense of stigma from mental health professionals and society in general which appeared to be enhanced by simultaneous condition of living with BD into older adulthood.

In general, the group reported an increase in lower mood which is consistent with previous research indicating older adults with BD present more predominantly with depressive symptoms[12 13] and milder episodes of mania, consistent with previous research.[14] Laidlaw *et al*[30] highlight that depression is not an automatic outcome of old age or an inevitable response to challenges of ageing. Therefore, developing an understanding for the predominance of depressive symptoms for older people with BD is important.

There were differences in the group's concept of recovery and whether the term was useful to describe their experience of living and coping with BD over the life span, based on whether they saw this as a clinical or personal concept. A study exploring the concept of recovery in older people versus younger people with mental health problems found that the older generation was not aiming towards a new or revised sense of identity.[31] However, those who identified themselves as managing their difficulties competently felt they had sustained or recovered their sense of self. Findings from the current study support this as individuals who held positive views about their recovery journey and were engaging in meaningful activities, were the ones who felt their self-identity and confidence was now intact.

While the group appeared positive about the development of an intervention for this population, one member questioned how other older people may perceive it. Research has found that older adults with depression do hold positive opinions about psychological therapy and if offered a choice would prefer them over psychotropic medication (however they may be less likely to be offered or receive this[32]).

### Considerations for clinical practice

The themes generated from the focus groups have led to a number of important recommendations for clinicians when delivering RfT for older people with BD. There were consistencies with younger individuals' priorities for recovery, where hope has been highlighted as a key part of the recovery process.[33] The group had clear ideas about what they would like from a therapist and wanted to work on areas of personal growth such as building assertiveness, confidence and competence. These targets identified with older people align with findings from Jones *et al*'s[15] RfT trial with a younger cohort where improvements in personal recovery were associated with improvements in personal growth and self-esteem.[34] The group identified a number of adaptions they thought would be helpful for RfT for older people, based on their experience of living with BD in later life. These were consistent with guidelines developed for working with an older adult client group.[30] There was a strong sense of the need to be valued and respected for their 'survival' of

---

### Box 1    Considerations for delivering RfT for older people

**Recovery stance**
► Explore the person's concept of recovery (traditional vs personal recovery).[4]
► Identify whether the word feels acceptable to use or find another to describe.[4]
► Identify and build on any pre-existing strategies which have helped the person progress on their recovery journey.[4]
► Use 'hope' as a key message for helping a person progress towards their recovery goals.[4]

**Symptom management**
► Identify specific health-related and age-related changes which may be impacting a person's current presentation.[1]
► Explore whether there has been a change in symptoms over time (eg, more depressive state now) and how they have coped with this.[2]
► Be aware that the consequence of repeated episodes over time (and limited life span) may have led to a greater sense of hopelessness for change.[2]
► Allow time to explore issues such a guilt and shame in later life and the impact on mood episodes now.[2]
► Explore the person's experience of receiving care from both relatives and health professionals and how this has affected their attitudes towards new opportunities for support.[3]

**Specific adaptations**
► Session length from 50 min to an hour.[6]
► To enhance memory and learning; use strategies such as repetition and association and write summaries at the end of each session (revisiting these at the beginning of the next session).[6]
► Use images, film or audio recordings as a way of enhancing a person's therapeutic experience.[6]
► Use clear and simple language and making study materials accessible and visually interesting.[6]
► Consider the use of booster sessions at the end of therapy to revisit strategies and enhance outcomes.[6]

**Therapist values**
► Spend time building a relationship, exploring a longer and potentially more complex history, allowing time at the beginning and the end of the session.[5]
► Treat the older person with dignity and respect, develop a shared understanding of the wealth of experience the person with BD brings to the therapeutic relationship (challenging any perceived stigma the person may identify with related to living with BD as an older person).[5]
► Provide transparency and acknowledge any differences between the therapist and the client with regard to age, gender, ethnicity and the therapist's lack of BD diagnosis.[5]

**Therapist focus**
► Nurture pre-existing strengths and acknowledge the resilience already present of living a life with a long-term condition.[5]
► Work on building assertiveness, confidence and competence to manage bipolar-related experiences, maintaining an encouraging stance throughout.[5]

*1=health-related and age-related changes, 2=experience of BD in later life, 3=managing and coping with BD in later life, 4=experience of recovery in later life, 5=seeking help in the future and 6=adapting RfT for older people.

BD, bipolar disorder; RfT, recovery-focused therapy' in box 1.

BD and the challenges which have presented throughout their lifespan. Knight and Laidlaw[35] identify 'wisdom' as a useful frame of reference when identifying and developing goals working with older depressed people. Therefore, building on a person's 'wisdom' and life skills learnt in the face of adversity feels like an important part of the therapeutic process with this client group.

This is the first study to involve older people with BD to shape and develop an intervention for their cohort, enhancing the quality, value and relevance of the recommendations. The recommendations have been used in the second phase of the programme of work which has evaluated RfT for older people in a randomised controlled trial.[16] They also provide a helpful framework for clinicians working with older people with BD in wider clinical practice, based on the service user's priorities. The recommendations are summarised in box 1 and mapped onto the six themes identified in the results section.

### Strengths and limitations

The service user researcher cofacilitated the first group with the aim of facilitating open and honest discussions. They were able to identify with a lot of the issues raised by the group and their presence in the first group may have had some impact on some of the discussions.

When drawing any conclusions from the data, it is evident that there are a number of methodological limitations which must be addressed. The individuals with lived experience taking part in the group are not representative of all older people with BD. They were a small, self-diagnosed group of people from the North West of England who had an active interest in taking part in the study. They were all white British and all had (or had retired from) a professional working background.

Finally, the study employed a careful analysis to explore patterns in the individual's accounts of experience. However, all studies that are based on self-report are constrained by the context and also subject to recall bias.

### CONCLUSIONS

The study aimed to explore a number of topics relevant to living with BD in later life and use this information to enhance the pre-existing RfT for older adults. In general, the groups were positive about the development of a therapeutic approach for this specific cohort. They were able to give insight into the realities of living with BD across the life span. The groups highlighted a number of challenges that they had faced; however, nurturing the resilience and wisdom developed as a product of coping with adverse circumstances appears to be a key part of what they want from the therapeutic process.

**Acknowledgements** The authors would like to thank all of the individuals who took part in the study.

**Contributors** All authors contributed to development and preparation of the study design and topic guide. ET and RL conducted the focus groups. ET led the analysis, with contributions from FL and SHJ. ET wrote the draft of the manuscript, which was proofed, edited and approved by FL, SHJ and RL.

**Funding** This study was funded by the National Institute for Health Research through the Doctoral Research Fellowship Programme, Grant No DRF-2014-07-094.

**Competing interests** None declared.

**Patient consent for publication** Not required.

**Ethics approval** The study was approved by the UK National Health Service Ethics Committee process (REC ref: 15/NW/0330). All participants were asked to provide written consent prior to the focus groups.

**Provenance and peer review** Not commissioned; externally peer reviewed.

**Data availability statement** No data are available. The transcripts cannot be published or made available if requested to protect the anonymity of the participants who took part in the focus groups. There are additional quotes from the focus groups in online supplemental file 2.

**ORCID iD**
Elizabeth Tyler http://orcid.org/0000-0001-5955-5607

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
