## [Reviewer comments · BMJ Open]

ARTICLE DETAILS

TITLE (PROVISIONAL)	Developing a recovery-focused therapy for older people with bipolar disorder: A qualitative focus group study
AUTHORS	Tyler, Elizabeth; Lobban, Fiona; Long, Rita; Jones, Steven

VERSION 1 – REVIEW

REVIEWER	Wendy Jones Greater Manchester Mental Health NHS Foundation Trust, Psychosis Research Unit
REVIEW RETURNED	12-May-2021

GENERAL COMMENTS	This is an interesting paper on an important topic. Methods are appropriate and qualitative work is an asset to the larger study of RfCBT. It is positive that it has a strong PPI focus and that a service user researcher was involved. It acknowledges the limitations of the sample and the reflexivity section is good. The quotes are relevant and thought-provoking for each of the themes. However, the following spelling/grammar errors were spotted: Line 64 - should say 'of' rather than 'if'. Line 106-107 should say 'consented to being approached'. Line 129 should say 'researcher was not able to attend', Line 181 should say 'an' inductive approach, Line 183 should say "The original topic", Line 184 should say "seeking help in the future", Line 229 - remove the apostrophe at the end of the word "don't", Line 535 - should say "comparison", Line 530 needs a space between 1. and the author surname, and the reference on Line 534 doesn't need to say (2019) twice.
---

REVIEWER	Kazuki Matsumoto Chiba University
REVIEW RETURNED	16-May-2021

GENERAL COMMENTS	Overall comment As the number of chronic mental illnesses increases in an aging society with longer lifespans, it is very meaningful to study the needs of older people with bipolar disorder and consider preferred services. This research theme is novel and the manuscript contains important insights for the general reader. However, this manuscript has not been properly reviewed for previous researches at the introduction. In addition, the description of research projects that are not related to the research purpose is described in the introduction, and the explanation until the research question is derived is insufficient. On the other hand, the research methods carried out are accurate, the results are well organized, and the consideration and conclusions of the results are clear. Therefore, I believe that with a major revision of the introduction, this manuscript will be able to publish the research results as a manuscript that is easier for the reader to understand. The result of my peer review is that I recommend that you revise the
--

	manuscript according to the comments below. Comment 1 Title It is strange that it is is the word 'therapy' after CBT. Comment 2 Abstract Objective: The purpose of this study has nothing to do with RCTs. Isn't the purpose of extracting considerations when providing CBT for the elderly with BD? Comment 3 Introduction Please provided your review of previous studies to investigate the clinical features and needs of older people with bipolar disorder. Comment 4 I don't think RCTs fit the purpose of this study. It is understandable that the research project is constructed in two stages. However, since this manuscript reports research results limited to the first stage, please clearly describe only the first stage. My suggestion is to add an introduction to the significance of doing the first stage research and to remove the description of RCTs from this manuscript. Comment 5 Methods Sampling and recruitment Did you have a target number of people to study, including the parties, family, and friends? Did you consider gender? If there is any agreement or research plan that you have pointed out before the research, please disclose it in the revised manuscript. Coment 6 Please provide a definition of "elderly" in this study. The study involves a 48-year-old party. In most CBT studies, older people are at least 50 years old (Gould RG et al., 2012. https://agsjournals.onlinelibrary.wiley.com/doi/full/10.1111/j.1532-5415.2012.04166.x). Coment 7 Results Theme 1: Health and age-related change in later life Please discuss the interpretation of the results obtained on discussion in your paper. Please exclude the following sentences. "Consistent with the older adult literature" at results. Line 200 to 201. Page 12. Coment 8 Discussion In discussion, consider how the developed RfCBT presented in Table 4 has been improved over the past and what positive therapeutic impacts it can have on the elderly.
--	---

VERSION 1 – AUTHOR RESPONSE

Reviewer 1:

Line 64 - should say 'of' rather than 'if'. Line 106-107 should say 'consented to being approached'. Line 129 should say 'researcher was not able to attend', Line 181 should say 'an' inductive approach, Line 183 should say "The original topic", Line 184 should say "seeking help in the future", Line 229 - remove the apostrophe at the end of the word "don't", Line 535 - should say "comparison", Line 530 needs a space between 1. and the author surname, and the reference on Line 534 doesn't need to say (2019) twice

Response

The manuscript has been carefully reviewed and all of the editing changes indicated by Reviewer 1 have been made in the revised manuscript.

Reviewer 2:

Comment 1:

It is strange that it is the word 'therapy' after CBT.

Response

The word CBT has been removed from the title.

Comment 2:

Objective: The purpose of this study has nothing to do with RCTs. Isn't the purpose of extracting considerations when providing CBT for the elderly with BD?

The reference to the RCT has been removed from the abstract, please see line 25-26 on the tracked changes document.

Comment 3:

Introduction

Please provided your review of previous studies to investigate the clinical features and needs of older people with bipolar disorder.

Response

A review of previous studies to investigate the clinical features and needs of older people with bipolar has been included. Please see lines 73-83 on the tracked changes document.

Comment 4:

I don't think RCTs fit the purpose of this study. It is understandable that the research project is constructed in two stages. However, since this manuscript reports research results limited to the first stage, please clearly describe only the first stage. My suggestion is to add an introduction to the

significance of doing the first stage research and to remove the description of RCTs from this manuscript.

Response

The description of the RCT has been removed from the introduction. The aims of the first stage have been more clearly articulated in the introduction, please see lines 86-100.

Comment 5:

Methods

Sampling and recruitment

Did you have a target number of people to study, including the parties, family, and friends? Did you consider gender? If there is any agreement or research plan that you have pointed out before the research, please disclose it in the revised manuscript.

Response

The sampling and recruitment section has been broadened to describe the recruitment aims further. Please see lines 126-129.

Comment 6:

Please provide a definition of "elderly" in this study. The study involves a 48-year-old party. In most CBT studies, older people are at least 50 years old (Gould RG et al., 2012.

<https://eur02.safelinks.protection.outlook.com/?url=https%3A%2F%2Fagsjournals.onlinelibrary.wiley.com%2Fdoi%2Ffull%2F10.1111%2Fj.1532-5415.2012.04166.x&data=04%7C01%7Cjones7%40live.lancs.ac.uk%7C3e20034b90234927856d08d925e7b7ba%7C9c9bcd11977a4e9ca9a0bc734090164a%7C0%7C0%7C637582499431627263%7CUnknown%7CTWFpbGZsb3d8eyJWljojMC4wLjAwMDAiLCJQIjoiV2luMzliLCJBTiI6Ikk1haWwiLCJXVCi6Mn0%3D%7C1000&data=hU4f7%2BKGlx0NgzFI1RJXEGK%2FzocRI3LVtTz%2BbJJhzTw%3D&reserved=0>

Response

All participants who were service users with bipolar disorder are over 60 years old. The 48-year-old person was a friend/carer of an older person with bipolar disorder. This has been further clarified by separating out the participants' characteristics for those with lived experience of bipolar over the age of 60 from carer/friend participants, please see table 3 and line 183-193.

Comment 7:

Results

Theme 1: Health and age-related change in later life Please discuss the interpretation of the results obtained on discussion in your paper. Please exclude the following sentences. "Consistent with the older adult literature" at results. Line 200 to 201. Page 12.

Response

The sentence "Consistent with the older adult literature" has been removed from the results section and there has been further interpretation in the discussion section of theme 1. Please see lines 424-428.

Comment 8:

Discussion

In discussion, consider how the developed RfCBT presented in Table 4 has been improved over the past and what positive therapeutic impacts it can have on the elderly.

Response

A discussion about how recommendations have been developed with individuals living with the condition and how this improves recovery focused therapy for older people has been further elaborated. Please see the “Considerations for clinical practice section”, lines 458 – 477.

VERSION 2 – REVIEW

REVIEWER	Kazuki Matsumoto Chiba University
REVIEW RETURNED	03-Jul-2021
GENERAL COMMENTS	The authors' revised manuscript fully responds to my comments. The revised manuscript contains new recommendations for the treatment of bipolar disorder in the elderly and is novel.